# MicroRNA-27b-3p Targets the Myostatin Gene to Regulate Myoblast Proliferation and Is Involved in Myoblast Differentiation

**DOI:** 10.3390/cells10020423

**Published:** 2021-02-17

**Authors:** Genxi Zhang, Mingliang He, Pengfei Wu, Xinchao Zhang, Kaizhi Zhou, Tingting Li, Tao Zhang, Kaizhou Xie, Guojun Dai, Jinyu Wang

**Affiliations:** 1College of Animal Science and Technology, Yangzhou University, Yangzhou 225009, China; 13013706620@163.com (M.H.); Wu_P_Fei@163.com (P.W.); zpf_xc@163.com (X.Z.); zkz282816@163.com (K.Z.); d150070@yzu.edu.cn (T.L.); zhangt@yzu.edu.cn (T.Z.); yzxkz168@163.com (K.X.); daigj@yzu.edu.cn (G.D.); jywang@yzu.edu.cn (J.W.); 2Joint International Research Laboratory of Agriculture & Agri-Product Safety, Yangzhou University, Yangzhou 225009, China

**Keywords:** miR-27b-3p, *MSTN*, myoblast, proliferation, differentiation

## Abstract

microRNAs play an important role in the growth and development of chicken embryos, including the regulation of skeletal muscle genesis, myoblast proliferation, differentiation, and apoptosis. Our previous RNA-seq studies showed that microRNA-27b-3p (miR-27b-3p) might play an important role in regulating the proliferation and differentiation of chicken primary myoblasts (CPMs). However, the mechanism of miR-27b-3p regulating the proliferation and differentiation of CPMs is still unclear. In this study, the results showed that miR-27b-3p significantly promoted the proliferation of CPMs and inhibited the differentiation of CPMs. Then, myostatin (*MSTN*) was confirmed to be the target gene of miR-27b-3p by double luciferase reporter assay, RT-qPCR, and Western blot. By overexpressing and interfering with *MSTN* expression in CPMs, the results showed that overexpression of *MSTN* significantly inhibited the proliferation and differentiation of CPMs. In contrast, interference of *MSTN* expression had the opposite effect. This study showed that miR-27b-3p could promote the proliferation of CPMs by targeting *MSTN*. Interestingly, both miR-27b-3p and *MSTN* can inhibit the differentiation of CPMs. These results provide a theoretical basis for further understanding the function of miR-27b-3p in chicken and revealing its regulation mechanism on chicken muscle growth.

## 1. Introduction

In recent years, with more and more attention paid to microRNA (miRNA), it has become the focus of biological research. With the early discovery of miRNA let-7 in nematodes, researchers have gradually found that miRNAs are widely distributed in animals and plants [1]. Moreover, miRNA is a class of endogenous noncoding single-chain small molecule RNA, about 21–26 nucleotides [2]. It cannot encode proteins, but it can participate in post-transcriptional regulation. Furthermore, miRNAs can degrade or inhibit mRNA by matching with mRNA of protein-encoding genes, resulting in the mRNA level and translation efficiency decreased [3]. Additionally, miRNA has a wide range of biological functions and participates in various cell processes, including proliferation, differentiation, apoptosis, metabolism, invasion, metastasis, and drug resistance [4,5]. Besides this, the expression of many miRNAs is tissue-specific, such as miR-1, miR-133, miR-206, and so on, and these miRNAs specifically expressed in muscle tissue are called myogenic miRNAs (myomiRs) [6,7,8]. In the process of skeletal muscle differentiation, the regulation of genes in the myomiRs family is essential. However, many non-myomiRs family genes also play a key role, which indicates that miRNAs play an important role in skeletal muscle growth and development [9,10].

It was established that miR-27b-3p belongs to the miRNA precursor family of miR-27, consisting of two members, miR-27a and miR-27b, located on chromosomes 8 and 13, respectively [11,12]. Previous studies have shown that miR-27b-3p plays a regulatory role in many kinds of cancer cells [13,14,15]. Recent studies have shown that overexpression of miR-27b-3p could inhibit the proliferation of C2C12 mouse myoblasts, while inhibition of miR-27b-3p could promote the differentiation of C2C12 mouse myoblasts [16]. However, the role of miR-27b-3p in the proliferation and differentiation of chicken primary myoblasts is still unclear.

Based on the target genes prediction of miR-27b-3p and RNA-seq results, we identified myostatin (*MSTN*) as a candidate target gene of miR-27b-3p. *MSTN*, also known as growth differentiation factor 8 (*GDF-8*), is a member of transforming growth factor-β (*TGF-β*), which plays an important role in the regulation of skeletal muscle growth [17]. *MSTN* plays a negative regulatory role in skeletal muscle development and mammalian growth [18]. Studies have shown that disruption of *MSTN* signaling can promote muscle growth in cattle, and the mRNA expression level of the *MSTN* gene in longissimus dorsi of low-birth-weight piglets is 65% higher than that of normal piglets [19,20]. Studies showed that the administration of myostatin inhibited the differentiation of satellite cells derived from the pectoralis major (PM) muscles and had no effect on satellite cells from biceps femoris (BF). However, administration of anti-myostatin antibodies did not affect the differentiation of either PM or BF cells [21].

The purpose of this study was to explore the effects of miR-27b-3p and its target gene, *MSTN*, on the proliferation and differentiation of chicken primary myoblasts (CPMs), and to clarify the regulatory mechanism between them, to help understand the molecular mechanism of miRNAs and related genes in chicken skeletal muscle growth and development, and provide the theoretical basis for chicken molecular breeding.

## 2. Materials and Methods

### 2.1. Ethics Statement

All animal experimental protocols in this study were carried out in strict accordance with the “Jiangsu Province laboratory animal management measures”, and approved by the Animal Ethics Committee of Yangzhou University (Approval number:Yzu DWLL-201903-001). All efforts were made to minimize animal suffering.

### 2.2. Experimental Animals and Tissues

The eggs of Haiyang Yellow Chickens’ first female parent, Jinghai Yellow Chickens, were provided by Jiangsu Jinghai Poultry Industry Group Co., Ltd. (Nantong, China), and hatched in Jiangsu Genetics and Breeding Laboratory. Four female Jinghai Yellow Chickens at the age of 12 embryonic days (E12), E14, E16, E18, E20, and at 1 day old (day 1) (the sex was judged by the development of the gonad) were randomly selected. At the same time, four female Jinghai Yellow Chickens at the age of 4 weeks old (4W), 8W, 12W, and 16W were randomly selected. The chest and leg muscle tissues from the five embryonic stages (E12–E20) and four growth stages (4W–16W) were collected. Meanwhile, a total of ten types of tissue (the heart, liver, spleen, lung, kidney, intestine, stomach, brain, chest muscle, and leg muscle) were collected from the 1-day-old female chickens. These tissues were collected and frozen in the −80 °C refrigerator, in the RNA preservation solution (Vazyme, Nanjing, China), for subsequent RNA extraction. 

### 2.3. Cell Culture

HEK293T cells: HEK293T cells were cultured in DMEM High-Glucose medium (Sigma-Aldrich, St. Louis, MO, USA) containing 10% fetal bovine serum (Gibco, Grand Island, NY, USA) and 1% penicillin–streptomycin–amphotericin (Solarbio, Beijing, China).

DF-1 cells: DF-1 cells were cultured in DME-F12 (Sigma-Aldrich, St. Louis, MO, USA) medium containing 10% fetal bovine serum and 1% penicillin–streptomycin–amphotericin.

Isolation and culture of chicken primary myoblasts: The leg muscles of 11/12-day-old chick embryos were taken. The bones and blood clots in the muscles were separated, and the muscles were evenly cut into a 10 cm cell culture dish. The muscle was digested with collagenase type I (Gibco, Grand Island, NY, USA), at 37 °C, to obtain single cells. The cell suspension after digestion was filtered and centrifuged, and then cultured in a 10 cm cell culture dish. Fibroblasts were removed after three times differential attachment. The growth medium was DME-F12, containing 20% fetal bovine serum and 1% penicillin–streptomycin–amphotericin. The differentiation medium was DME-F12 containing 4% fetal bovine serum and 1% penicillin–streptomycin–amphotericin. All cells were cultured in a cell incubator (Binder, Tuttlingen, Germany), with 5% CO_2_, at 37 °C.

### 2.4. Total RNA Extraction and Quality Detection, Complementary DNA (cDNA) Synthesis, and Quantitative Real-Time PCR (qRT-PCR)

Total RNA was extracted from tissues and cells by Trizol (Takara, Dalian, China) and stored at −80 °C. The integrity of RNA was detected by agarose gel electrophoresis, and the quality and concentration of RNA were detected by spectrophotometer (Thermo, Waltham, MA, USA). Moreover, miRNA cDNA was synthesized by using miRNA 1st strand cDNA synthesis Kit (by stem-loop) (Vazyme, Nanjing, China). Then the expression of microRNA-27b-3p in tissues and cells was detected by miRNA Universal SYBR qPCR Master Mix (Vazyme, Nanjing, China). The cDNA synthesis of genes was performed, using Hiscript QRT Supermax (Vazyme, Nanjing, China). The expression levels of related genes in tissues and cells were detected, using the ChamQ SYBR qPCR Master Mix (Vazyme, Nanjing, China). QRT-PCR was performed in the Applied Biosystems 7500 Fast DX real-time PCR instrument (ABI, Los Angeles, California, USA), using chicken *β-actin* and *U6* as reference genes. Three repeated tests were performed on each sample. The relative expressions of miR-27b-3p and related genes were calculated by the 2^−ΔΔCT^ method [22].

### 2.5. Primers for Quantitative Real-Time PCR (qRT-PCR)

For the miR-27b-3p and *U6*, primers were designed, using V1 software (Vazyme, Nanjing, China), combined with the stem–loop method. For the genes, primers were designed by using Premier Primer 5.0 software (Premier Biosoft International, Palo Alto, CA, USA). Meanwhile, the primer sequences for the *p21*, myosin heavy chain (*MYHC*), myogenic determination 1 (*MYOD1*), and myogenin (*MYOG*) genes refer to the sequences in the work of Cai et al. [23]. These primers were synthesized by Sangon Biotech (Shanghai, China) and are shown in Table 1 and Table 2.

### 2.6. RNA Oligonucleotides and Plasmids Construction

The miR-27b-3p mimic, mimic-NC (negative control), miR-27b-3p inhibitor, inhibitor NC, small interfering RNAs (siRNAs) used for knockdown of *MSTN*, and non-specific siRNA negative control were designed and synthesized by GenePharma (Suzhou, China). The oligonucleotide sequences are shown in Table 3.

For the overexpression of *MSTN*, a full-length coding domain sequence (CDS) of the *MSTN* gene was synthesized by GenePharma (Shanghai, China). The synthesized sequence was inserted into the pcDNA-3.1 vector (Promega, Madison, WI, USA). Moreover, the restriction sites were BamHI and EcoRI.

For the construction of the double luciferase reporter vector, the 3′ UTR region containing the predicted binding site of miR-27b-3p and *MSTN* was amplified by PCR, and the amplified fragment was cloned into PMIR double luciferase reporter vector, according to the homologous recombination method. The restriction sites were HindIII and mIuI. Moreover, the constructed plasmid was sequenced to check whether the target fragment was inserted successfully. The Fast Mutagenesis Kit V2 (Vazyme, Nanjing, China) was used to construct the mutation vector, according to the instructions. The predicted binding sites were successfully mutated from ACTTGAA to CGCCTGC for the PMIR-MSTN-3′UTR-MT vector. Primers used for vector construction are shown in Table 4.

### 2.7. Cell Transfection

All cell transfection procedures followed the jetPRIME Transfection Reagent (polyplus, Illkirch, France), following the manufacturer’s protocol.

### 2.8. Dual-Luciferase Reporter Assay

HEK293T cells and DF-1 cells were used in the double luciferase assay. Firstly, HEK293T cells and DF-1 cells were evenly seeded in a 24-well cell culture dish. When the cells reached the appropriate density, they were co-transfected with PMIR-MSTN-3′ UTR-WT (wild type) or PMIR-MSTN-3′ UTR-MT (mutant) and miR-27b-3p mimic or mimic NC. After 48 h of transfection, the luciferase activity was detected on the multi-mode micropore detection system (EnSpire, Perkin Elmer, USA), according to the instructions of the double luciferase detection kit (Vazyme, Nanjing, China).

### 2.9. CCK-8 Assay

The CPMs were seeded in 96-well cell culture dishes, with an added growth medium. When the cells reached the appropriate density, the cells were transfected. According to the instruction manual of CCK-8 Kit (Vazyme, Nanjing, China), cell proliferation was detected at 12, 24, 36, and 48 h after transfection, and the absorbance at 450 nm was detected by the multi-mode micropore detection system (EnSpire, Perkin Elmer, USA).

### 2.10. EdU Assay

The CPMs were seeded in 24-well cell culture dishes with an added growth medium. When the cells reached the appropriate density, the cells were transfected. After 48 h of transfection, the cells were washed with 1 × PBS (Solarbio, Beijing, China) and fixed with 4% paraformaldehyde (Solarbio, Beijing, China) for 30 min. Then the cells were treated according to the instructions of the EdU Apollo In Vitro Imaging Kit (RiboBio, Guangzhou, China). Finally, three regions were randomly selected, using a fluorescence inverted microscope (DMi8, Leica, Germany) to evaluate the number of stained cells. Image-Pro software was used for data analysis.

### 2.11. Cell Cycle was Detected by Flow Cytometry

The CPMs were seeded in 6-well cell culture dishes supplemented with a growth medium. When the cells reached the appropriate density, the cells were transfected. After 48 h of transfection, the cells were collected and placed in 2 mL enzyme-free tubes. 70% ethanol was added to each tube, to fix the cells overnight, at −20 °C. Then, the cells were stained with propidium iodide (50 ug/mL, Solarbio, Beijing, China) containing RNaseA (50 ug/mL, TianGen, Beijing, China) and incubated in the dark, at 37 °C, for 30 min. Finally, the cells were analyzed on the FACSAria SORP flow cytometer (BD company, Franklin, NJ, USA). ModFit LT software was used for data analysis.

### 2.12. Immunofluorescence

The CPMs were seeded in 12-well cell culture dishes supplemented with a growth medium. When the cells reached the appropriate density, the cells were transfected. After 12 h of transfection, the growth medium was replaced by a differentiation medium, to induce the differentiation of CPMs. After 72 h of differentiation, the cells were washed with PBS (Solarbio, Beijing, China) and fixed with 4% paraformaldehyde (Solarbio, Beijing, China) for 30 min. After the cells were fixed, the cells were permeated with 0.5% Triton X-100 (Solarbio, Beijing, China) for 15 min, and then 10% goat serum (Solarbio, Beijing, China) was added and incubated for 60 min, at 37 °C. At the end of incubation, goat serum was removed, and the first antibody (MYHC, Proteintech, Wuhan, China, 1:500) was added. Then the cells added with the first antibody were incubated in the dark at, 4 °C, overnight. After the first antibody was incubated, the cells were washed with 1 × PBST (Solarbio, Beijing, China), and the second antibody (FITC-IgG, Bioss, Beijing, China, 1:400) was added and incubated, at 37 °C, for 1 h. After the second antibody was incubated, the second antibody was aspirated, and the cells were cleaned with 1 × PBST. Then 4’,6-diamidino-2-phenylindole (DAPI, Beyotime, Shanghai, China) staining solution was added. After incubation at room temperature for 5 min, DAPI was aspirated, and cells were cleaned with 1 × PBST. Finally, an anti-fluorescence quenching agent (Beyotime, Shanghai, China) was added into each well, and the cells were observed and photographed by a fluorescence inverted microscope (DMi8, Leica, Germany). The experimental data were analyzed with Image-Pro Plus 6.0. We calculated the percentage of myotube coverage area in the total area as the myotube area.

### 2.13. Western Blot Assay

Western and IP cell lysates (Beyotime, Shanghai, China) were added to CPMs. Then, the lysate cells were centrifuged at 12,000 rpm, for 10 min, and the supernatant was collected. After determining the protein concentration with BCA Protein Quantification Kit (Vazyme, Nanjing, China), the protein samples were polyacrylamide gel electrophoresis (GenScript, Nanjing, China) and then transferred to PVDF (BIO-RAD, Hercules, CA, USA). The PVDF membrane was sealed with 4% skimmed milk powder, at room temperature, for 1 h, and then the primary antibody was added and incubated overnight, at 4 °C. Then, the PVDF membrane was washed with 1 × TBST (Solarbio, Beijing, China) and incubated with a secondary antibody, at room temperature, for 1 h. Electrochemiluminescence (ECL) was used for imprinting display. Moreover, the relative expression of the protein was obtained by ChemDoc^TM^Touch Imaging System (Bio-Rad, Hercules, CA, USA). The antibody and its dilution ratio were as follows: MYHC rabbit polyclonal antibody (Proteintech, Wuhan, China, 1:500), MSTN rabbit polyclonal antibody (Bioss, Beijing, China, 1:500), GAPDH rabbit polyclonal antibody (HUABIO, Hangzhou, China, 1:500), and HRP binding Goat anti-rabbit IgG (BBI, Shanghai, China, 1:5000).

### 2.14. Statistical Analysis

Statistical analysis was performed by using SPSS18.0 software (SPSS Inc., Chicago, IL, USA). The unpaired Student’s *t*-test was used for two-group comparison analysis. A one-way ANOVA was used for multiple-group comparison analysis. Duncan’s multiple range test was used to determine the significance. The data were considered statistically significant when *p* < 0.05 (*) or *p* < 0.01 (**). Each experiment was repeated three times, and all data are presented as least squares means ± SEM (standard error of the mean).

## 3. Results

### 3.1. Expression of miR-27b-3p and MSTN in Tissues of Chicken

Firstly, the expression of miR-27b-3p in different tissues of one-day-old Jinghai Yellow Chickens was detected by qRT-PCR. The results showed that the expression of miR-27b-3p in chest and leg muscles was higher than that in other tissues (Figure 1a). Meanwhile, during the development of the chicken embryos, the expression of miR-27b-3p in chest and leg muscles continued to increase (Figure 1c). Although the expression of miR-27b-3p in chest muscle was stable, the expression of miR-27b-3p in leg muscle increased from 4 weeks old to 12 weeks old (Figure 1d). All of these indicate that miR-27b-3p is involved in the development of skeletal muscle of chicken. Moreover, the expression of *MSTN* in different tissues of one-day-old Jinghai Yellow Chickens was detected by qRT-PCR. The results showed that the expression of *MSTN* in chest and leg muscles was higher than that in other tissues (Figure 1b). Simultaneously, during the development of the chicken embryos, the expression of *MSTN* in chest and leg muscles continued to increase from 12 to 18 embryo age and then decreased continuously (Figure 1e). Moreover, the expression of *MSTN* in chest and leg muscles was decreased continuously from 4 to 16 weeks old (Figure 1f). These suggest that *MSTN* may be involved in the development of chicken skeletal muscle in both embryonic and growth stages.

### 3.2. MiR-27b-3p Promotes CPMs Proliferation

To understand the role of miR-27b-3p in the proliferation of CPMs, overexpression and inhibition experiments of miR-27b-3p in CPMs were conducted. The results showed that overexpression of miR-27b-3p significantly inhibited the expression of *p21* mRNA, reduced the number of G0/G1 phase cells, and significantly increased S-phase cells (Figure 2a,c and Appendix A). On the contrary, inhibition of miR-27b-3p significantly promoted the expression of *p21*, resulting in an increase in the number of G0/G1 phase cells and a decrease in the number of S-phase cells (Figure 2b,d and Appendix A). Moreover, CCK-8 and EdU assays showed that overexpression of miR-27b-3p promoted CPMs proliferation (Figure 2e,g,i), while inhibition of miR-27b-3p inhibited CPMs proliferation (Figure 2f,h,j). These results indicate that miR-27b-3p can promote the proliferation of CPMs.

### 3.3. MiR-27b-3p Inhibits CPMs Differentiation

To understand the effect of miR-27b-3p on CPMs differentiation, CPMs were induced to differentiate in vitro. The results showed that the expression level of miR-27b-3p in the process of differentiation was lower than that in the proliferation stage, which indicated that miR-27b-3p was involved in CPMs differentiation (Figure 3a). Then, after overexpression and knockdown of miR-27b-3p in CPMs, the mRNA expression levels of *MyoD*, *MyoG* and, *MyHC*, and the protein expression level of *MyHC* were detected. Compared with the control group, the mRNA expression levels of three differentiation marker genes and the protein expression level of *MyHC* in CPMs transfected with miR-27b-3p mimic were significantly lower (Figure 3b,d). In contrast, transfection of miR-27b-3p inhibitor significantly promoted the mRNA expression of three differentiation marker genes and the protein expression of *MyHC* (Figure 3c,e). Moreover, an miR-27b-3p mimic and inhibitor were transfected into CPMs, and the myotube differentiation was detected by immunofluorescence staining. The results showed that overexpression of miR-27b-3p significantly inhibited CPMs’ differentiation and reduced total myotube area (Figure 3f,h), while the knockdown of miR-27b-3p significantly promoted CPMs differentiation and increased total myotube area (Figure 3g,i). These results suggest that miR-27b-3p can inhibit the differentiation of CPMs.

### 3.4. MSTN Is a Target Gene of miR-27b-3p

To understand the molecular mechanism of miR-27b-3p regulating gene expression, we found that the seed sequence in miR-27b-3p is conserved among vertebrates (Figure 4a). Then, its target gene was predicted using TargetScan Human 7.2 online software, and the results showed that the seed sequence of miR-27b-3p could completely bind to sites 80–86 in the 3′UTR region of the *MSTN* gene (Figure 4c). Moreover, the potential interaction model from RNAhybrid showed that the binding site was in a stable format (Figure 4b). These indicated that *MSTN* might be a target gene of miR-27b-3p. To verify whether the *MSTN* gene is the target gene of miR-27b-3p, a double luciferase reporter assay was carried out in HEK293T cells and DF-1 cells (Figure 4d,e). We transfected the recombinant vectors (PMIR-MSTN-3′UTR-WT and PMIR-MSTN-3′UTR-MT) and miR-27b-3p mimic or mimic-NC (normal control) into the two kinds of cells, respectively. Compared to the group with mimic-NC, the results showed that the luciferase activity was significantly reduced after transfection with miR-27b-3p mimic in the wild-type recombinant vector (PMIR-MSTN-3′UTR-WT) group. Meanwhile, there was no difference in the mutant recombinant vector group (PMIR-MSTN-3′UTR-MT). Besides, after overexpression of miR-27b-3p in the proliferative myoblasts, the mRNA and protein expression levels of *MSTN* in CPMs were significantly decreased (Figure 4f,h). At the same time, inhibition of miR-27b-3p resulted in up-regulation of *MSTN* mRNA and protein expression (Figure 4g,i). Simultaneously, the rescue experiment showed that co-transfection of miR-27b-3p mimic and pcDNA 3.1-MSTN in the proliferative phase cells could reduce the proliferation-promoting effect of miR-27b-3p mimic (Figure 4j). These results indicate that *MSTN* is the target gene of miR-27b-3p.

### 3.5. The Facilitation of miR-27b-3p on CPMs Proliferation was Achieved by Its Target Gene MSTN

The above results showed that *MSTN* is the target gene of miR-27b-3p. To understand the role of the *MSTN* gene in the proliferation of CPMs, we constructed the overexpression vector of the *MSTN* gene and synthesized the interference sequence of *MSTN* (Figure 5a,b). Through Western blot assay, the results showed that overexpression of *MSTN* could increase the protein expression level of *MSTN* in CPMs. At the same time, interference with *MSTN* can reduce the expression level (Figure 5c,d). This indicates that the constructed overexpression vector and the synthesized interference sequence can be used for subsequent overexpression and interference experiments. In CPMs, the results showed that overexpression of *MSTN* significantly increased the expression of the *p21* gene, reduced the number of S-phase cells, and increased the number of G0/G1 phase cells (Figure 5e,g and Appendix A). However, inhibition of *MSTN* expression resulted in the opposite effect (Figure 5f,h and Appendix A). CCK-8 and EdU assays showed that the overexpression of the *MSTN* gene significantly inhibited CPMs proliferation (Figure 5i,k,m), while the inhibition of the *MSTN* gene expression significantly promoted CPMs’ proliferation (Figure 5j,l,n). These results suggest that the facilitation of miR-27b-3p on CPMs’ proliferation was achieved by its target gene, *MSTN*.

### 3.6. The Inhibition of miR-27b-3p on MSTN Is Different between CPMs Proliferation and Differentiation

We have confirmed that *MSTN* is the target gene of miR-27b-3p. The CPMs were induced to differentiate in vitro when they had reached 70–80% confluency. In the process of inducing CPMs the differentiation in vitro, the results showed that the expression level of *MSTN* in the differentiation phase was lower than that in proliferation stage (Figure 6a). This expression trend is similar to miR-27b-3p, which can directly inhibit *MSTN* expression. To understand the effect of *MSTN* on CPMs differentiation, we detected the mRNA expression levels of *MyoD*, *MyoG*, and *MyHC* after overexpression and knockdown of *MSTN* in CPMs. We also detected the protein expression level of *MyHC* by Western blot. Compared with the control group, the mRNA expression of three differentiation markers and the protein expression of *MyHC* were significantly decreased in CPMs transfected with pcDNA 3.1-MSTN (Figure 6b,d). On the contrary, siR-MSTN transfection significantly promoted the expression of three differentiation marker genes and increased the protein expression level of *MyHC* (Figure 6c,e). Besides this, pcDNA 3.1MSTN and siR-MSTN were transfected into CPMs, and the myotube differentiation was detected by immunofluorescence staining. The results showed that the overexpression of *MSTN* significantly inhibited myoblast differentiation and reduced the total myotube area (Figure 6f,h). At the same time, the knockdown of *MSTN* significantly promoted myoblast differentiation and increased the total myotube area (Figure 6g,i). These results suggest that miR-27b-3p cannot restrict *MSTN* expression and function during CPMs differentiation. To further validate these results, the mRNA expression of *MSTN* after transfection of miR-27b-3p mimic in the differentiation stage of CPMs was also detected. The results showed that miR-27b-3p inhibited the expression of *MSTN* in the proliferation phase, but there was no significant difference in the expression levels of *MSTN* in the differentiation phase (Figure 6j). These results suggest that the inhibition of miR-27b-3p on *MSTN* is different between CPMs’ proliferation and differentiation.

## 4. Discussion

This study revealed the role of miR-27b-3p in the proliferation and differentiation of CPMs. Based on the fact that miR-27b-3p can target *MSTN*, we further studied the regulatory role of the *MSTN* gene in the proliferation and differentiation of chicken CPMs. Compared with the early embryonic stage, Diana et al. [24] detected many miRNAs in three primary germ layers (endoderm, ectoderm, and mesoderm) of chickens, which indicated that miRNAs played an important role in regulating organogenesis, differentiation, and post-differentiation events. Many studies have shown that miR-27b-3p is involved in the growth and development of skeletal muscle and the occurrence of diseases [25,26,27]. In our previous RNA-seq, we found that miR-27b-3p was differentially expressed during myoblast proliferation and differentiation [28]. In this study, we studied the expression of miR-27b-3p in chicken skeletal muscle during embryonic development and growth stages. We found that miR-27b-3p was up-regulated in 12-embryonic-age to 20-embryonic-age and 1-day-old chest and leg muscles, which indicated that miR-27b-3p might play an important role in skeletal muscle development. Besides this, we found that the expression of miR-27b-3p in the differentiation phase was significantly lower than that in the proliferation phase, which was consistent with the previous sequencing data. At the same time, Cai et al. [23] found that miR-16-5p inhibited the differentiation of primary myoblasts. The expression level of miR-16-5p in the differentiation phase was significantly lower than that in proliferation phase, which was consistent with our conclusion. All of these indicate that it plays a regulatory role in the proliferation and differentiation of chicken primary myoblasts. To further understand the regulatory role of miR-27b-3p in skeletal muscle growth, we conducted a series of experiments in vitro. We found that miR-27b-3p in CPMs could promote their proliferation and inhibit their differentiation.

Moreover, miRNAs regulate the expression of a target gene by binding to the 3 ‘UTR region of target mRNA [29]. Therefore, identifying target genes is very important for the functional identification of miRNAs [30]. In many cancer cells, the function of miR-27b-3p and its target genes has been studied [31,32,33]. In this study, we predicted the target gene of miR-27b-3p by TargetScan software and found that the seed sequence of miR-27b-3p can bind to the 3′UTR region of the *MSTN* gene. Then, we conducted double luciferase experiments in HEK293T cells and DF-1 cells and found that the overexpression of miR-27b-3p significantly reduced luciferase activity. At the same time, after the overexpression of miR-27b-3p in CPMs, which were in the proliferative phase, we found that the mRNA and protein expression levels of *MSTN* were decreased. In contrast, inhibition of miR-27b-3p had the opposite effect. Besides this, the rescue experiments showed that the overexpression of miR-27b-3p and *MSTN* in CPMs could down-regulate the role of miR-27b-3p in proliferation. These results indicate that *MSTN* is the target gene of miR-27b-3p, and miR-27b-3p regulates CPMs proliferation by targeting *MSTN*.

As an important part of the muscle, muscle fiber density affects muscle protein content. In broiler growth and development, the number of muscle fibers is mainly determined in the embryonic stage [34]. At the same time, myostatin (*MSTN*) is an important gene regulating embryonic muscle cells and adult muscle development, so *MSTN* is crucial for skeletal muscle growth and development [35]. Studies have shown that recombinant myostatin protein can inhibit C2C12 mouse myoblasts proliferation, DNA synthesis, and protein synthesis [36]. Another study showed that overexpression of *MSTN* cDNA in C2C12 cells could down-regulate the mRNA levels of *MyoD*, *MYOG*, and the activity of downstream targeting creatine kinase, thereby reversibly inhibiting myogenesis [37]. In this study, we found that the expression of *MSTN* in chest and leg muscles from 12 to 18 embryonic age increased continuously during embryonic development, reached the peak at 18 embryonic age, and then decreased continuously. Meanwhile, we found that the expression of *MSTN* in chest and leg muscles that were from 4 to 16 weeks old showed a downward trend during growing stages. The tissue expression profile analysis of one-day-old chickens showed that *MSTN* was highly expressed in chest muscle and leg muscle, which indicated that *MSTN* might play an important role in muscle growth and development. In vitro, we found that overexpression of *MSTN* significantly inhibited the proliferation of CPMs. Both *MyoD* and *MyoG* play an important role in the regulation of myoblasts differentiation. *MyoD* acts as a myogenic determination gene, while *MyoG* is essential for the differentiation of myoblasts [38,39]. Moreover, the myosin heavy chain (*MyHC*) isoform is considered to be the main determinant of the fast and slow types of muscle fibers. It is a molecular marker for distinguishing muscle fiber types and studying muscle fitness [40]. Therefore, we used them as differentiation marker genes in this study. We found that overexpression of *MSTN* decreased the expression of *MyoG*, *MyoD*, and *MyHC* and inhibited the differentiation of CPMs. Here, we report that *MSTN* can inhibit the proliferation and differentiation of CPMs, and miR-27b-3p can inhibit the expression of *MSTN* in proliferative CPMs (Figure 4f,h). However, it should be noted that miR-27b-3p only inhibited the expression of *MSTN* in the proliferative CPMs, and the inhibitory effect of miR-27b-3p on *MSTN* was significantly reduced in the differentiation phase (Figure 6j). It has been found that a single miRNA can bind to multiple target genes, and the inhibitory effect of miRNA on these target genes varies with the functional characteristics of the target sites [41]. Therefore, the inhibitory effect of miR-27b-3p on the differentiation of CPMs may be regulated through other pathways.

## 5. Conclusions

In conclusion, this study reported that the proliferation and differentiation of CPMs were regulated by miR-27b-3p and *MSTN*. MiR-27b-3p could inhibit the expression of *MSTN* in the proliferative CPMs and promote the proliferation of CPMs. Both miR-27b-3p and *MSTN* could inhibit the differentiation of CPMs.

## Figures and Tables

**Figure 1 cells-10-00423-f001:**
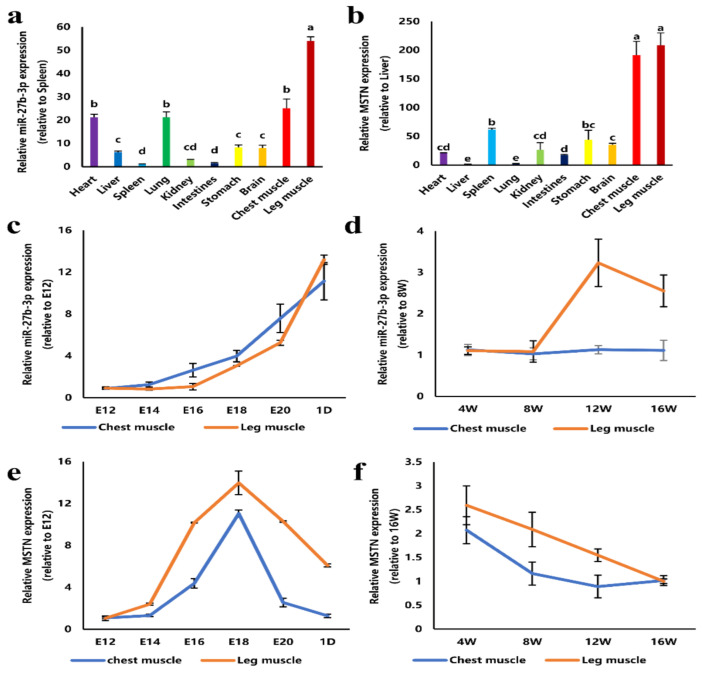
Expression of miR-27b-3p and myostatin (*MSTN*) in different tissues of Jinghai Yellow Chickens. (**a**) The expression of miR-27b-3p in tissues of Jinghai Yellow Chickens. (**b**) The expression of *MSTN* in tissues of Jinghai Yellow Chickens. (**c**) Expression of miR-27b-3p in the chest and leg muscles at the embryonic stage and oneday old. (**d**) Expression of miR-27b-3p in the chest and leg muscles of Jinghai Yellow Chickens at 4, 8, 12, and 16 weeks old. (**e**) Expression of *MSTN* in the chest and leg muscles at the embryonic stage and one day old. (**f**) Expression of *MSTN* in the chest and leg muscles of Jinghai Yellow Chickens at 4, 8, 12, and 16 weeks old. A one-way ANOVA was used for multiple-group comparison analysis. Duncan’s multiple range test was used to determine the significance. The same lowercase letters indicate no significant difference between the two groups, while different lowercase letters indicate a significant difference between the two groups. Data are presented as mean ± SEM (standard error of the mean) (*n* = 4).

**Figure 2 cells-10-00423-f002:**
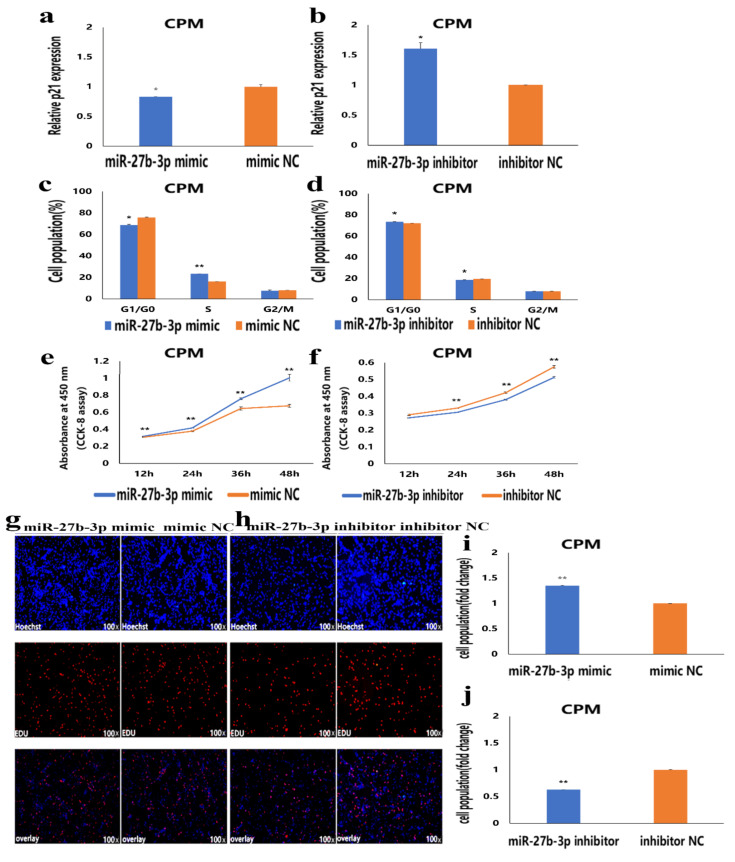
The proliferation of chicken primary myoblasts (CPMs) is promoted by miR-27b-3p. (**a**,**b**) The relative mRNA expression of *p21* after transfection with miR-27b-3p mimic and inhibitor in CPMs. (**c**,**d**) Cell-cycle analysis of CPMs 48 h after transfection with miR-27b-3p mimic and inhibitor in CPMs, using propidium iodide staining for DNA content. (**e**,**f**) Cell growth was measured following the transfection of miR-27b-3p mimic and inhibitor in CPMs. (**g**,**h**) The proliferation of CPMs following the transfection with miR-27b-3p mimic and inhibitor was assessed by EdU incorporation. Microscopic images were obtained by a fluorescence inverted microscope (dmi8, Leica, Germany). (**i**,**j**) Proliferation rates of CPMs following transfection with miR-27b-3p mimic and inhibitor. In all graphs, the results are shown as mean ± SEM (standard error of the mean) (*n* = 3). Statistical significance of differences between means was assessed, using the unpaired Student’s *t*-test (* *p* < 0.05; ** *p* < 0.01) vs. NC (negative control).

**Figure 3 cells-10-00423-f003:**
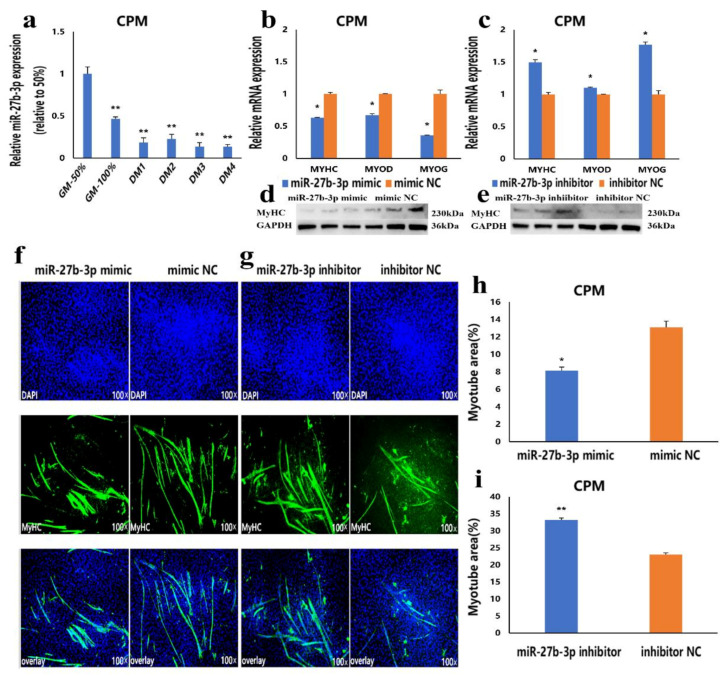
CPMs’ differentiation is inhibited by miR-27b-3p. (**a**) The relative expression of miR-27b-3p during CPMs’ proliferation and differentiation (GM 50% and GM 100% represent the proliferation density of 50% and 100%; DM1-DM 4 represents differentiation from one to four days). (**b**,**c**) The mRNA expression levels of three differentiation marker genes after transfection with miR-27b-3p mimic and inhibitor in CPMs. (**d**,**e**) The protein expression level of *MyHC* after transfection with miR-27b-3p mimic and inhibitor in CPMs. (**f**,**g**) MyHC staining of CPMs at 72 h after transfection with miR-27b-3p mimic and inhibitor in CPMs. Microscopic images were obtained by a fluorescence inverted microscope (dmi8, Leica, Germany). (**h**,**i**) Myotube area (%) of CPMs 72 h after overexpression and inhibition of miR-27b-3p. In all graphs, the results are shown as mean ± SEM (standard error of the mean) (n = 3). Statistical significance of differences between means was assessed, using the unpaired Student’s *t*-test (* *p* < 0.05; ** *p* < 0.01) vs. NC (negative control).

**Figure 4 cells-10-00423-f004:**
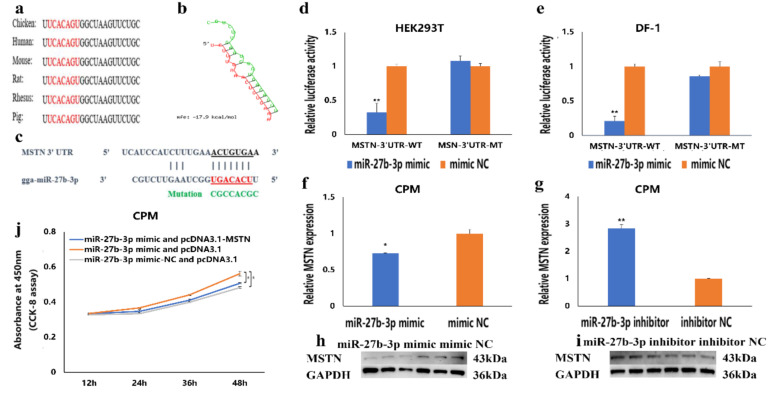
*MSTN* is the target gene of miR-27b-3p. (**a**) The seed region of miR-27b-3p. (**b**) The potential interaction model between miR-27b-3p and *MSTN* 3′UTR from RNAhybrid. (**c**) The potential binding site sequence of miR-27b-3p on *MSTN* 3′UTR. The seed sequences and mutant sequences were highlighted in red and green, respectively. (**d**,**e**) The luciferase assays were performed by the co-transfection of wild-type or mutant *MSTN* 3′ UTR with a miR-27b-3p mimic or mimic-NC in HEK293T and DF-1 cells. (**f**–**i**) After transfection of miR-27b-3p mimics, miR-27b-3p inhibitor, or NC, the expression of *MSTN* was determined by qRT-PCR and Western Blot. (**j**) CCK-8 detection in rescue experiment. In all graphs, the results are shown as mean ± S.E.M (standard error of the mean) (*n* = 3). Statistical significance of differences between means was assessed using the unpaired Student’s *t*-test (* *p* < 0.05; ** *p* < 0.01) vs. NC (negative control).

**Figure 5 cells-10-00423-f005:**
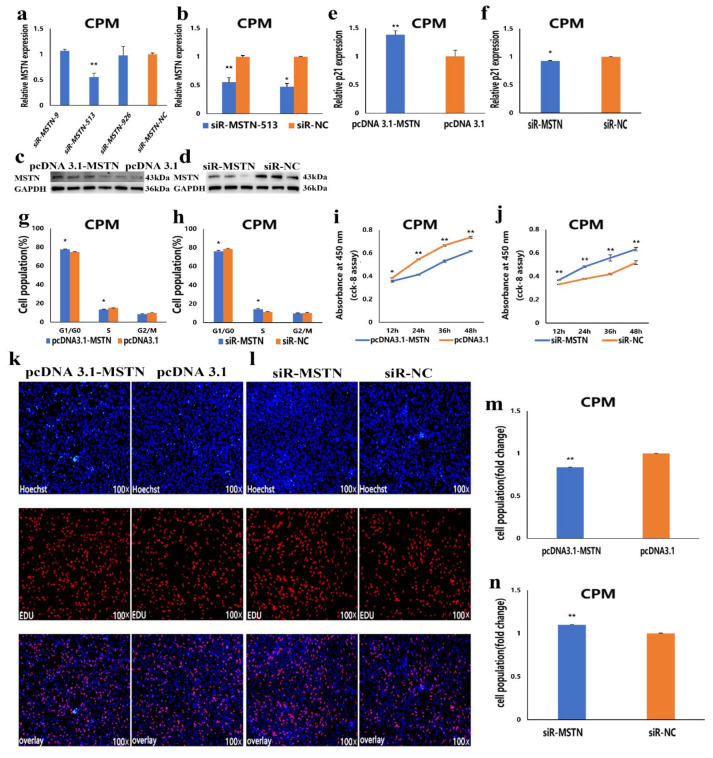
The facilitation of miR-27b-3p on CPMs proliferation was achieved by its target gene *MSTN*. (**a**,**b**) Screening of *MSTN* gene interference sequence and detection of interference efficiency. (**c**,**d**) The protein expression levels of *MSTN* in CPMs after overexpression and interference. (**e**,**f**) The relative mRNA expression of *p21* after transfection with pcDNA 3.1-MSTN and pcDNA 3.1 in CPMs. (**g**,**h**) Cell-cycle analysis of CPMs 48 h after transfection with pcDNA 3.1-MSTN and pcDNA 3.1 in CPMs, using propidium iodide staining for DNA content. (**i**,**j**) Cell growth was measured following the transfection of pcDNA 3.1-MSTN and pcDNA 3.1 in CPMs. (**k**,**l**) The proliferation of CPMs following transfection with pcDNA 3.1-MSTN and pcDNA 3.1 was assessed by EdU incorporation. Microscopic images were obtained by a fluorescence inverted microscope (dmi8, Leica, Germany). (**m**,**n**) Proliferation rates of CPMs following transfection with pcDNA 3.1-MSTN and pcDNA 3.1. In all graphs, the results are shown as mean ± SEM (standard error of the mean) (*n* = 3). Statistical significance of differences between means was assessed, using the unpaired Student’s *t*-test (* *p* < 0.05; ** *p* < 0.01) vs. NC (negative control).

**Figure 6 cells-10-00423-f006:**
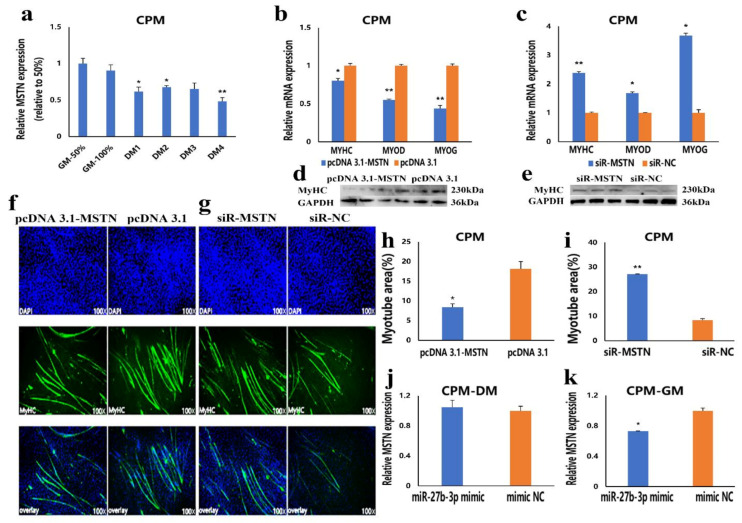
The inhibition of miR-27b-3p on *MSTN* is different between CPMs proliferation and differentiation. (**a**) The relative expression of *MSTN* during CPMs’ proliferation and differentiation (GM 50% and GM 100% represent the proliferation density of 50% and 100%; DM1-DM 4 represents differentiation from one to four days. (**b**,**c**) The mRNA expression levels of differentiation marker genes after transfection with pcDNA 3.1-MSTN and siR-MSTN in CPMs. (**d**,**e**) The protein expression levels of *MyHC* after transfection with pcDNA 3.1-MSTN and siR-MSTN in CPMs. (**f**,**g**) MyHC staining of CPMs at 72 h after transfection with pcDNA 3.1-MSTN and siR-MSTN in CPMs. Microscopic images were obtained by a fluorescence inverted microscope (dmi8, Leica, Germany). (**h**,**i**) Myotube area (%) of CPMs 72 h after overexpression and inhibition of *MSTN*. (**j**,**k**) The proliferating (GM) and differentiating (DM) CPMs were transfected with miR-27b-3p mimic, respectively, and the relative *MSTN* mRNA expression levels were then analyzed. In all graphs, the results are shown as mean ± SEM (standard error of the mean) (*n* = 3). Statistical significance of differences between means was assessed, using the unpaired Student’s *t*-test (* *p* < 0.05; ** *p* < 0.01) vs. NC (negative control).

**Table 1 cells-10-00423-t001:** miRNA primers used for qRT-PCR.

Gene	Primer Sequence (5′-3′)	Annealing Temperature (°C)
Gga-miR-27b-3p	F:GCGCGTTCACAGTGGCTAAG	60
	R:AGTGCAGGGTCCGAGGTATT	
Stem-loop primer	GTCGTATCCAGTGCAGGGTCCGAGG	
	TATTCGCACTGGATACGACGCAGAA	
*U6*	F:GTCACTTCTGGTGGCGGTAA	60
	R:GTTCAGTAGAGGGTCAAA	

**Table 2 cells-10-00423-t002:** Gene primers used for qRT-PCR.

Gene	Primer Sequence (5′-3′)	Product Size (bp)	Annealing Temperature (°C)	Accession Number
*MSTN*	F:GCTTTTACCCAAAGCTCCTCCAC	179	60	NM_001001461.1
	R:AGCAACATTTTGGTTTTCCCTCC			
*P21*	F:GAGATGCTGAAGGAGATCAATGAG	102	60	NM_204396.1
	R:GTGGTCAGTCCGAGCCTTTT			
*MYOD1*	F:GCTACTACACGGAATCACCAAAT	200	60	NM_204214.2
	R:CTGGGCTCCACTGTCACTCA			
*MYHC*	F:CTCCTCACGCTTTGGTAA	213	60	NM_001319304.1
*MYOG*	R:TGATAGTCGTATGGGTTGGT	320	60	NM_204184.1
	F:CGGAGGCTGAAGAAGGTGAA			
	R:CGGTCCTCTGCCTGGTCAT			
*β-actin*	F:CAGCCATCTTTCTTGGGTAT	169	60	NM_205518.1
	R:CTGTGATCTCCTTCTGCATCC			

**Table 3 cells-10-00423-t003:** Oligonucleotide sequences for overexpression and inhibition.

Fragment Name	Sequence (5′-3′)
miR-27b-3p mimic	UUCACAGUGGCUAAGUUCUGC
	AGAACUUAGCCACUGUGAAUU
mimic-NC	UUCUCCGAACGUGUCACGUTT
	ACGUGACACGUUCGGAGAATT
miR-27b-3p inhibitor	GCAGAACUUAGCCACUGUGAA
inhibitor-NC	CAGUACUUUUGUGUAGUACAA
siR-MSTN	GCAGAUCCUGAGACUCAUUTT
	AAUGAGUCUCAGGAUCUGCTT
siR-NC	UUCUCCGAACGUGUCACGUTT
	ACGUGACACGUUCGGAGAATT

**Table 4 cells-10-00423-t004:** Primers used for vector construction.

Primer Name	Primer Sequence (5′-3′)	Product Size (bp)	Annealing Temperature (°C)
MSTN-3′ UTR-WT	F:AAAGATCCTTTATTAAGCTTCTGTCGTGAGATCCACCATT	298	62
	R:ATAGGCCGGCATAGACGCGTCCCATTTGTTAAATCCGGTG		
MSTN-3′ UTR-MT	F:TTGAACGCCACGCATTACGTACGCTAGGCATTGCC	6742	68
	R: GTAATGCGTGGCGTTCAAAGATGGATGAGGGGATATAG

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
