# Peer review of "MicroRNA-27b-3p Targets the Myostatin Gene to Regulate Myoblast Proliferation and Is Involved in Myoblast Differentiation"

_cells, 2021, doi:10.3390/cells10020423_

Round 1
Reviewer 1 Report
In this study, authors investigated the role of miR-27b-3p and MSTN in proliferation and differentiation of CPM.
Major remarks:
Page 2, line 54 – regarding the reference 17, and 18 authors should cite the original work and not their own work, in which the original work was previously cited.
Figure 1 – What was the control sample to normalize the values? For all figure panels.
Figure 2a and 2b – to what was the sample normalized for the fold change step?
Page 8, figure 4d, e – Can the transfection be done on CPM cells and can authors comment on that in the text.
Figure 4j- can authors comment is the rescue effect due to scavenging of the miR-27b-3p through pcDNA3.1-MSTN or overexpression of MSTN itself?
Minor remarks:
In the headline instead of abbreviation, full name myostatin should be used.
page 1 lane 6 – delete “and”
page 1, line 33 – replace “no-encoding” with “noncoding or non-coding”
page 1, line 39 – reference missing
page 2, line 48 - what are the C2C12 cells?
Page 5 – Figure 2c and 2d – letters are shifter compared to “a” and “b”
Figure 3a – is differentiation induced at 50% or 100% growth? If at 100% growth, one does not exclude that decrease in miR-27b-3p expression could be due to contact inhibition and not differentiation.
Figure 3b and 2c – what sample was used as control to normalize and get the relative expression?
Page 7 – line 172, HEK293T?
Figure 2e – since in the rest of the figures, blue color always represents treatment, only in this one treatment is orange color. Can it be synchronized for the better overview?
Instead of miR-inhibitor, anit-MIR term should be used through the text.
Page 2, line 77 – either to use term “breast” of “chest” muscle. In the Figure 1 both are used “Chest muscle” and “breast muscle” which I assume should be the same.
Reviewer 2 Report
In this paper the Authors report data concerning the involvement of miR-27b-3p in the regulation of the growth and differentiation of chicken myoblasts, through the reduction of its direct target gene myostatin.
The work is very preliminary and needs of further experiments. For instance, exploring the involvement of potential upstream and downstream pathways; or analyzing the expression of cyclins, fibronectin, caderins, ecc. It would also be interesting to evaluate the regulation of miR-27b-3p in different stages of growth. Moreover, what are the direct mechanisms that link the expression of MyHc to MSTN?
The work presents several issues regarding the grafts and the writing text. It is very difficult to understand the figures. In all figures, the measure units are not reported and it is not clear what is the reference of the relative expression.
For istance, in figure 1A data referred to breast are not reported and the significance method is not clear.
It would be necessary to use different colors for the molecules used and keep them (see Figure 2E F and Figure 5I J where the colors are inverted).
The images in Fig.2 G H, Fig. 3 F G and Fig 5 K L are too small. It would be better to increase the magnification, so the figures are more easy to understand.
The images of the cell cycle should be reported and not just the histograms relating to the phases of the cycle.
Decide: p21 or P21!
Sometimes the significance shown in the figure does not reflect the phenomenon seen.
In the materials and methods, W.B section, the antibodies against MSTN and MyHC are reported as GDF8 and MYH1 respectively. It would be better to use the same in the results. Please correct miR-27b-5p on page 7 line 173.
Round 2
Reviewer 2 Report
The manuscript has been revised in accordance with the Reviewer's suggestions
Author Response
Thank you very much for all the suggestions for this article, which have improved the manuscript. We have further revised the manuscript. Thank you again for your work!
This manuscript is a resubmission of an earlier submission. The following is a list of the peer review reports and author responses from that submission.